# Effectiveness of Educational Interventions to Increase Skills in Evidence-Based Practice among Nurses: The EDITcare Systematic Review

**DOI:** 10.3390/healthcare10112204

**Published:** 2022-11-02

**Authors:** Omar Portela Dos Santos, Pauline Melly, Roger Hilfiker, Katia Giacomino, Elodie Perruchoud, Henk Verloo, Filipa Pereira

**Affiliations:** 1Department of Nursing Sciences, School of Health Sciences, HES-SO Valais/Wallis, University of Applied Sciences and Arts Western Switzerland, Chemin de l’Agasse 5, CH-1950 Sion, Switzerland; 2Institute of Health Sciences, Universidade Católica Portuguesa, Rua de Diogo Botelho 1327, 4169-005 Porto, Portugal; 3Department of Physiotherapy, School of Health Sciences, HES-SO Valais/Wallis, University of Applied Sciences and Arts Western Switzerland, Rathausstrasse 8, CH-3954 Leukerbad, Switzerland; 4Service of Old Age Psychiatry, Department of Psychiatry, Lausanne University Hospital, Route de Cery 60, CH-1008 Lausanne, Switzerland

**Keywords:** evidence-based practice, primary healthcare, beliefs, knowledge, implementation, nurses, interventions, education

## Abstract

Background: Using evidence-based practice (EBP) improves the implementation of safe, high-quality healthcare for patients, reduces avoidable costs, and plays a crucial role in bridging knowledge–action gaps and reducing health inequities. EBP combines the best available evidence in the relevant literature with patient preferences and values and healthcare professionals’ (HCPs) expertise. Methods: Systematic searches of ten bibliographic databases, unpublished works, and the Grey Literature Report sought studies published up to 30 September 2022. Results: The 15 studies retained involved 2712 nurses. Three types of effective educational interventions were identified: (1) multifaceted educational strategies incorporating mentoring and tutoring; (2) single educational strategies, often delivered online; and (3) multifaceted educational strategies using the five steps of EBP. Eleven primary outcomes (EBP beliefs, EBP self-efficacy, perceived EBP implementation, EBP competencies, EBP knowledge, EBP skills, EBP attitudes, EBP behaviors, EBP desire, EBP practice, and perceptions of organizational culture and readiness) were assessed using 13 qualitative and quantitative instruments. Conclusions: Ensuring the successful implementation of EBP requires effective educational strategies. Computer-based learning seems the most cost-effective and efficient strategy, when considering caregivers’ characteristics, the clinical field, and educational interventions across the pre-, peri-, and post-implementation processes.

## 1. Introduction

The reports “To Err is Human” (1999) [1] and “Crossing the Quality Chasm” (2001) [2] created a level of global awareness that pushed healthcare institutions to take corrective actions to promote and improve the quality of clinical care and patient safety [3]. According to the Institute of Medicine, quality of care is “the ability of health services to increase the likelihood of achieving desired health outcomes in accordance with current professional knowledge” [3]. At the same time, the concept of patient safety—defined as the prevention of harm [4]—emerged as a major focus of professional vigilance. Healthcare systems today face the twin challenges of implementing EBP [5,6] to reduce healthcare costs [7] and increasing healthcare personnel’s job satisfaction [8]. Indeed, EBP contributes to improving the quality of care and patient safety because effectiveness, safety, responsiveness, efficiency, and equity have become indicators of whether care providers are using patient-centered practices. EBP is now an established framework among healthcare professionals (HCPs) [9,10,11,12]. As an integral part of patient-centered approaches, it involves the use of research steps to critically appraise research evidence and implement that evidence in practice [11,13,14]. In a nursing context, EBP is defined as the integration of information from various sources to aid clinical reasoning, including the best and most recent evidence, clinical expertise, personal experiences, patient preferences, and theories underpinning nursing care [5,6,15]. HCPs are expected to use EBP as a gold-standard approach to their daily practice [12,16,17,18,19,20]. One might expect EBP to be well known and understood by HCPs and, equally, to be prevalent in their education programs and clinical practice [15]. However, the routine use of EBP remains challenging [11] since between 30% and 40% of patients do not receive care based on current knowledge and two-thirds of good practice implementations fail [17,18,19]. Moreover, some HCPs feel that engaging in EBP is not within their scope of practice [20]. Because not all HCPs have undergone training on EBP along their career trajectory, there are questions about which educational interventions are most effective at increasing EBP skills among HCPs in their daily practice [12,21].

Nurses and physiotherapists are highly involved in healthcare services [22,23]. Because of their close relationships with healthcare users during their daily practice, they play important decision-making roles, strengthening communication and collaborative practices between community and specialized HCPs to provide the best available overall healthcare [24]. Although it is generally considered that nurses and physiotherapists—like every other HCP—are accountable for providing the best available evidence-based healthcare [25,26], recent research has concluded that only a tiny percentage of them consistently do so [13,27]. Indeed, multiple barriers to the daily use of EBP have been reported, such as inadequate skills in EBP among clinicians, time constraints, negative attitudes, a lack of personal motivation, and professional resistance to research [28,29,30,31,32]. Additionally, several authors have documented administrative and organizational problems in the workplace, a lack of mentors for EBP, inadequate point-of-care resources, gaps between theory and practice, a lack of any meaningful transition between training courses on EBP and the clinical reality, and absent or insufficient basic education on the subject [7,12,18,33,34]. It is therefore urgent that efficient and effective educational interventions that meet HCPs’ needs be identified and developed according to a well-defined implementation process that ensures that EBP’s added value is felt at the patient’s bedside. The importance of providing educational interventions on EBP has been recognized by the World Health Organization [33]. Over the last two decades, EBP in healthcare has been documented in exploratory and observational studies in different settings. Scurlock-Evans et al. [13] summarized attitudes, barriers, enablers, and EBP interventions among physiotherapists, although without specifying employment settings or assessing educational interventions. Melender et al. [34] summarized the educational interventions used to train nursing students to improve outcomes in implementing EBP. Nevertheless, to the best of our knowledge, no systematic reviews have examined the effectiveness of educational interventions to increase skills in EBP in the daily practice of nurses and physiotherapists. Furthermore, no previous international research has ever compared the outcomes of different educational interventions on EBP skills in our population of interest. While EBP is considered as a core competency needed for HCPs, evidence for how to effectively teach EBP remains suboptimal and there is an important heterogeneity of EBP education interventions. Therefore, there is a need to synthetize and evaluate the outcomes of the different EBP educational interventions proposed by the literature, and to evaluate what is the most effective to overcome the barriers cited, meet the characteristics and needs of the professionals, and make EBP a norm for al HCPs need to be identified. It is an essential process for bridging the gap between evidence and practice, for the nursing and physiotherapist professions to advance as well-informed disciplines, for the homogenization of basic education on EBP, and/or the avoidance of a lack of any meaningful transition between training courses on EBP and the clinical reality [35]. This review aimed to answer the questions: (a) What are the most effective educational interventions to increase skills in EBP among nurses and physiotherapists delivering primary healthcare? (b) What are the effects of these educational interventions on EBP skills among primary-care nurses and physiotherapists? We hypothesized that researchers, clinicians, healthcare system managers, healthcare institutions, healthcare policymakers, stakeholders, universities and their faculty, and healthcare professionals could identify the most effective educational interventions that would enable them to use evidence in daily practice, enable HCPs to understand EBP, promote and improve the quality of clinical care and patient safety, and improve responses to current healthcare challenges. If EBP educational interventions become an integral part of continuing HCPs education curricula, it will improve skills in EBP, facilitate its implementation, and guarantee its holistic benefits. Our second hypothesis was that the identification of these educational interventions would enable the recognition of the factors facilitating and constraining the implementation of EBP and would contribute to establishing effective implementation strategies within different healthcare settings.

## 2. Materials and Methods

This review was conducted following the Preferred Reporting Items for Systematic Reviews and Meta-Analyses (PRISMA) recommendations [36], the Meta-analysis Of Observational Studies in Epidemiology (MOOSE) reporting proposals [37], and the methods outlined in the Cochrane Handbook for Systematic Reviews of Interventions [38]. The research protocol for this systematic review has been published elsewhere [39].

### 2.1. Study Selection Criteria

#### 2.1.1. Participants and Healthcare Settings

This review considers studies involving registered HCPs delivering primary healthcare, including those with bachelor’s, master’s, and doctoral degrees in nursing (RNs = registered nurses) and physiotherapy (physiotherapists = PTs) and nursing and physiotherapy students in their respective educational institutions. Physical therapists and physiotherapists are considered synonymous. We also include all categories of primary healthcare settings for home-dwelling adult care, such as private offices, community, and health maintenance organizations (HMO), community and private primary healthcare settings, hospital outpatient departments (OPDs), and hospital settings.

#### 2.1.2. Types of Studies

Randomized controlled trials (RCTs), cluster randomized controlled trials (RCTs), non-randomized studies (NRSs), prospective cohort studies, case-control studies, controlled before-and-after studies, interrupted-time-series studies, quasi-experimental studies, and mixed-methods studies were included [40,41,42].

#### 2.1.3. Types of Educational Interventions

All kinds of educational interventions to improve the EBP skills of RNs, and PTs delivering active primary healthcare were examined. These included ex cathedra, interactive, online, or individual educational sessions on the steps and components of EBP, organized journal clubs, seminars, educational meetings, the distribution of educational materials, webinars and other individual-oriented educational activities, case studies, grand rounds, and mentoring within a healthcare organization. Educational interventions were excluded if they targeted the regulatory, economic, or financial aspects of EBP, as per the Cochrane Effective Practice and Organization of Care (EPOC, 2015) taxonomy of interventions [43].

#### 2.1.4. Outcome Measures

Our primary outcome was whether educational interventions increased or decreased the EBP knowledge, attitudes, beliefs, competencies, skills, self-efficacy, behavior, practice and perception of RNs, and PTs active in primary healthcare settings. This was measured using methods such as questionnaires, interviews, chart analyses, and self-reporting by RNs, and PTs, with a focus on dichotomous (yes/no), ordinal, or continuous scales or scores. Secondary outcomes explored the production of systematic reviews and numbers of journal clubs and grand rounds [44,45].

### 2.2. Search and Process Strategies

#### 2.2.1. Search Strategy and Identification of Relevant Studies

In collaboration with a medical librarian (PM) and using predefined search terms, we conducted a systematic literature search for published articles in the following electronic databases, from inception until 30 September 2022: Medline Ovid SP (from 1946), PubMed (NOT Medline[sb]) (from 1996), Embase.com (from 1947), CINAHL Ebesco (from 1937), the Cochrane Central Register of Controlled Trials Wiley (from 1992), PsycINFO Ovid SP (from 1806), Web of Science Core Collection (from 1900), PEDro (from 1999), the JBI Database of Systematic Reviews and Implementation Reports (from 1998), and the Trip Database (from 1997). We conducted a hand search of the bibliographies of all relevant articles and searched for unpublished studies using Google Scholar, ProQuest Dissertations and Theses dissemination, Mednar, and WorldCat. The search was completed a first time by exploring the grey literature in OpenGrey and the Grey Literature Report from inception until 31 December 2021, and a second time until 30 September 2022. We considered publications in any language.

#### 2.2.2. Study Screening and Data Extraction

Three reviewers (OPDS, HV, RH) independently screened the titles and abstracts identified in our searches to assess which studies met our inclusion criteria. The full texts of the articles retained were read to ensure the assessments were correct. Disagreements were resolved through discussion or, if needed, a consensus was reached after discussion with the co-authors (KG, EP, FP). A flowchart of the article selection process was drawn using the PRISMA statement guidelines [36] (Figure 1). Three reviewers (OPDS, EP, HV) extracted the data independently using a specially designed, standardized data-extraction form. Again, discrepancies were resolved through discussion and consultation with the co-authors if necessary. The information extracted from each study included: (1) study authors, year of publication, and country where the study was conducted; (2) study characteristics (including setting and design, duration of follow-up, and sample size); (3) participants’ basic sociodemographic characteristics; (4) characteristics of interventions (description and frequency of educational interventions, healthcare professionals involved, etc.); (5) characteristics of the usual care group; and (6) outcome measures.

### 2.3. Assessment of the Risks of Bias in the Studies Retained

Two reviewers (OPDS, HV) independently assessed the risks of bias in all the randomized and non-randomized studies of the interventions included. Disagreements were resolved through discussion and consultation with all the co-authors (EP, KG, RH, FP). Any disagreements in the quality assessments were resolved through discussion.

### 2.4. Statistical Analyses

Statistical analyses followed the recommendations of the Cochrane Handbook for Systematic Reviews of Interventions [38] and the PRISMA and MOOSE statements [36]. Descriptive statistics were used to describe the studies and participants involved. We computed the sum of the population and the mean age of the participants in the studies retained. We used exact statistics and measures of association to report on the effectiveness of the interventions in the studies. The extreme heterogeneity of the data collected meant that we could not compute group or subgroup meta-analyses.

### 2.5. Methodological Quality of the Studies Retained

Three different tools were used to assess the studies’ methodological quality. The Newcastle–Ottawa Scale (NOS) for assessing the quality of non-randomized studies in meta-analyses [47] was used to assess the only cohort study retrieved [48]. We used the validated Risk Of Bias In Non-randomized Studies of Interventions (ROBINS-I) tool [49] for the eight studies with a quasi-experimental design [50,51,52,53,54,55,56,57]. Finally, we used the Revised Cochrane Risk of Bias tool for randomized trials (RoB 2.0) [58]. Two researchers (OPDS, HV) independently rated the studies’ quality. Any disagreements about quality assessments were resolved by consensus discussion.

## 3. Results

### 3.1. Search Strategy and Results

Our literature search took place at two distinct time points, with a baseline search in 2019 and a supplementary search in December 2021: we retrieved a total of 18,299 references. The second search’s goal was to complete the first one and ensure up-to-date evidence. After removing duplicates, three researchers independently analyzed the titles and abstracts of 12,948 references. In the second phase, papers’ full texts were retrieved from the references and analyzed based on the inclusion and exclusion criteria. Finally, a third search was made in September 2022, during the revision process, with the objective of improving the literature review with information about the very latest research publications. Considering the importance of the topic we decided to conduct living systematic review based on the Cochrane recommendations, defined as “a systematic review that is continually updated, incorporating relevant new evidence as it becomes available” [59], by continual and active monitoring of the evidence, and incorporating relevant new evidence as it becomes available. Two appropriate new references were discovered. All the full texts respecting the inclusion criteria were analyzed and described in a structured reporting form (Figure 1).

### 3.2. Characteristics of the Retrieved Studies

The review included fifteen studies—nine in the United States of America [8,48,50,51,54,55,56,57,60], two in Taiwan [52,61], and one each in Portugal [62], Finland [63], Spain [53], and India [64]. All were published between 2011 and 2022. One study used a pre–post cohort design [48], four were RCTs [8,65,66], and ten used quasi-experimental designs [50,51,52,53,54,55,56,57,60,61]. Twelve studies were conducted in primary-care settings, one took place in a Portuguese nursing school [62], one in the nursing education institution of an Indian university [64], and one in Jaume University in Spain [53]. The total population included in the review was composed of 2712 RNs and licensed practical nurses (LPNs), bachelor’s degree students, Ph.D. students in Nursing Practice–Family Nurse Practice, nurse educators, emergency nurses, nurse managers, and visiting staff nurses. An original intention had been to look at PTs and primary HCPs but no relevant studies were found involving them.

The 1509 female and 108 male participants were aged between 19 and 65 years old. Seven studies did not report on their sample’s gender [50,54,55,56,61,64,66], and only nine studies provided information about their sample’s age distribution [48,50,51,52,53,60,61,62,64,66]. Table 1 describes the studies retained.

### 3.3. Methodological Quality of the Studies Retained

The methodological quality of the cohort study was poor (four stars) [48] (Table 2). The different domains of evaluation used in the quasi-experimental interventional studies also scored as moderate [50,51,52,53,54,55,56,57,60,61] (Table 3). Finally, the RCT studies were scored as “of some concern” regarding their methodological risk of bias [8,66,67,68,69] (Table 4).

### 3.4. Description of the Interventions

The studies retained had each developed a tailored educational intervention focusing on their sample’s specific needs. Those interventions were divided into three categories: (i) multifaceted educational strategies incorporating mentoring and tutoring [8,52,54,57,60,66,69], (ii) single educational strategies [55,58,59,61], and (iii) multifaceted educational strategies using the five steps (five As) of EBP [56,68,70].

**Table 1 healthcare-10-02204-t001:** Characteristics of the studies retained (*n* = 13).

Study	Authors	Year	Country	Professionals Involved	Study Design	Setting	Sample Mean Age (SD)	Men/Women	Covariates Included in the Study (Professional and Sociodemographic)
1	Cardoso, D., Couto, F., Cardoso, A. F., Bobrowicz-Campos, E., Santos, L., Rodrigues, R., Coutinho, V., Pinto, D., Ramis, M-A, Alves Rodrigues, M., and Apostolo, J. [62]	2021	Portugal	Students in the Bachelor of Nursing program, 8th semester	RCT cluster with two parallel brains qualitative section: - Evaluation of 18 bachelor’s degree students’ theses - IC *n* = 9 - CG *n* = 9	Portuguese nursing school	*n* = 11,148 - Mean age = 21.9 (SD = 2.2)	- Demographic characteristics - Effectiveness of an EBP educational program - Knowledge and skills in EBP - Qualitative evaluation of bachelor’s degree students’ theses	
2	Chao, W.-Y., Huang, L.C., Hung, H.-C., Hung, S.-C., Chuang, T.-F., Yeh, L.-Y., and Tseng, H.-C. [60]	2022	Taiwan	RNs with more than 3 months’ work experience	Quasi-experimental design - 114 licensed nurses - CG *n* = 54 - IC *n* = 54	475-bed regional teaching hospital	*n* = 114 - Mean age IC = 32.5 (SD = 8.84) - Mean age CG = 33.84 (SD = 7.53)	Women = 109 Men = 5	- Demographic characteristics - Knowledge, attitudes towards, and practice of EBP - Satisfaction score
3	D’Souza, P., George, A., Nair, S., Noronha, J., and Renjith, V. [63]	2021	India	Nurse educators involved in undergraduate and postgraduate nursing education programs	RCT	Nursing education institution in an Indian university	*n* = 51 - IG *n* = 27 - CG *n* = 24 - Mean age IG = 34.8 (SD = 6.2) - Mean age CG = 35.1 (SD = 7.6)	Not reported	- Effectiveness of an EBP training program - Knowledge, attitudes, usage, and competency - Demographic characteristics
4	Gallagher-Ford, L., Koshy Thomas, B., Connor, L., Sinnott, L. T., and Melnyk, B. M. [50]	2020	USA	Individuals who attended the EBP immersion program - 94.3% nurses - 5.7% non-nurses	Longitudinal pre-experimental design	Academic medical center (36.3%), health systems (25.5%), and community hospital (19.3%)	*n* = 400 - Mean age = 42.0 (SD = 11.0)	Not reported	- Effects of a 5-day EBP immersion program - EBP attributes, competencies, beliefs, implementation, and knowledge - Perceptions of organizational culture and readiness for integration of EBP - Demographic characteristics
5	Hart, P., Eaton, L.A., Buckner, M., Morrow, B. N., Barrett, D. T., Fraser, D. D., Hooks, D., and Sharrer, R. L. [51]	2008	USA	RNs and licensed practical nurses (LPNs)	A quasi-experimental design with a one-group pre–post-intervention survey design	Integrated healthcare system in a southeastern state in the United States	Pre-survey *n* = 744 Post-survey *n* = 314 - Mean age = 43.6 (SD = 10.3)	Women = 676 Men = 42	- Knowledge of EBP and research utilization - Attitudes and skills in EBP and research utilization
6	Hsieh, P.-L. and Chen, S.-H. [52]	2020	Taiwan	School nurses	Quasi-experimental, pre–posttest design	193 primary schools in Tayuan and New Taipei City	*n* = 401 - Mean age = 35.7 (min = 26, max = 58)	Women = 401 Men = 0	- Effectiveness of the multifaceted EBP training program - Knowledge, attitudes, skills, and self-efficacy
7	Koota, E., Kääriäinen, M., Kyngäs, H., Lääperi, M., and Melender, H.-L. [65]	2021	Finland	Emergency nurses	RCT with a parallel group	Four emergency departments at two university hospitals in Finland	*n* = 80 (IG = 40 and CG = 40) Completed study: *n* = 64 (IG = 34 and CG = 29)	Not reported	- Attitudes, knowledge, self-efficacy, skills, and behavior toward EBP - Satisfaction with the EBP educational intervention
8	Levin, R. F. Fineout-Overholt E., Melnyk, B. M., Barnes, M., and Vetter, M. J. [8]	2011	USA	Nurse managers and visiting staff nurses	Two-group randomized controlled pilot trial with a repeated-measures design	Community Health Services from the three boroughs of Queens, the Bronx, and Manhattan	*n* = 46 - IG *n* = 22 - CG *n* = 24	Women = 46 Men = 0	- Demographic characteristics - Nurses’ EBP beliefs and implementation behaviors - Group cohesion - Nurse job satisfaction - Nurse productivity - Nurse attrition/turnover rate - Manipulation checks on the intervention - Cost outcomes
9	Mena-Tudela, D., Gonzalez-Chorda. V.-M., Cervera-Gasch, A., Macia-Soler, M., and Orts-Cortés, M. I. [53]	2018	Spain	Second-year bachelor’s degree nursing students	Quasi-experimental before and after design	University Jaume I	*n* = 83 - Mean age = 21.6 (SD = 5.6)	Women = 70 Men = 13	- Knowledge, skills, and attitudes toward EBP - Demographic characteristics
10	Moore, L. [54]	2017	USA	RNs	Quasi-experimental, before and after design	RNs with a bedside practice in a healthcare facility	*n* = 77 - Pre-test *n* = 197 - Post-test *n* = 134	Not reported	- Effectiveness of an EBP educational intervention - EBP attitudes, knowledge, and skills - Correlation between educational preparation and years of nursing experience in using EBP, EBP attitudes, EBP knowledge, and EBP skills - Correlation between years of experience and using EBP, attitudes, knowledge, and skills - Demographic characteristics
11	Moore, L. K. [55]	2018	USA	RNs with bedside responsibilities	Quasi-experimental pre-test–post-test design	Regional hospital (nine counties in western Kentucky and two in southern Indiana)	*n* = 197	Not reported	- EBP skills - Practice, attitudes, knowledge, and skills related to EBP - Effect of educational preparation and years of nursing experience - RNs’ practice, attitudes, knowledge, and skills related to EBP - Demographic characteristics
12	Singleton, J.K. [48]	2017	USA	Doctor of Nursing Practice–Family Nurse Practice (DNP–FNP)	Pre–post cohort study	Primary care, hospital, nursing homes	*n* = 89 Mean age = 47.0 (min = 29, max = 65)	Women = 80 Men = 9	- Comparison between the five cohorts - Demographic characteristics
13	Underhill, M., Roper, K., Siefert, L. S., Boucher, J., and Berry, D. [56]	2015	USA	Oncology RNs and advanced practice nurses	Pre–post-test design	Ambulatory oncology setting, Dana-Farber Cancer Institute	*n* = 113	Not reported	- Barriers to involvement in EBP - Factors affecting participation in EBP projects - Feedback on how EBP can be better incorporated into nursing practice - Oncology nurses’ beliefs and implementation factors affecting EBP - Demographic factors associated with EBP beliefs and implementation
14	Waltz, L.-A., Munoz, L., Miller, R.-A., and Johnson, H.-W [61]	2022	USA	RNs employed full-time in an acute care setting at a metropolitan hospital	Quasi-experimental study with a pre-test–post-test design	179-bed metropolitan acute care hospital	*n* = 30 Age = 40–49 years (*n* = 9, 30.0%)	Not reported	- Demographic information - Self-perceptions of competencies in EBP understanding, ability, desire, and frequency - Self-perceptions of barriers before and after the intervention - Changes in knowledge and ability to use EBP
15	Wan, A. L. P. [57]	2017	USA	RNs working in medical/surgical/telemetry units, IMCU, ICU, or another unit (staff nurse, charge nurse, quality manager, nurse manager, nurse educator)	Pre-test–post-test quasi-experimental randomized design	Nurses in a single-site county hospital in the San Francisco Bay Area	*n* = 19 - IG *n* = 9 - CG *n* = 10 - Mean age EG = 40–49 (44.4%) - Mean age CG = 30–39 (55.6%)	Women = 9 Men = 9	- Demographic characteristics - Knowledge, beliefs, attitudes, and abilities in implementing EBP - Correlations between knowledge, beliefs, attitudes, and abilities in implementing EBP

Note: CG = control group; EBP = evidence-based practice; IG = interventional group; HCPs = healthcare professionals; ICU = intensive care unit; IG = intervention group; RNs = registered nurses.

**Table 2 healthcare-10-02204-t002:** Newcastle–Ottawa Scale (NOS) adapted for cohort studies.

	Selection				Comparability		Outcome			
Study	Representativeness of Exposed Cohort	Selection of Non-Exposed Cohort	Ascertainment of Exposure	Demonstration That Outcome of Interest Was Not Present at Start of the Study	Adjusted for the Most Important Risk	Adjusted for Other Risk Factors	Assessment of Outcome	Follow-Up Length	Loss-to-Follow-Up Rate	Total Quality
Singleton et al., 2017 [48]	1 *	0 *	1 *	1 *	0 *	0 *	0 *	1 *	0 *	4 *

Note: Studies were awarded a maximum of one star for each numbered item in the Selection and Outcome categories. A maximum of two stars could be awarded for Comparability. Studies were evaluated on a scale from 0 to 9 stars and classified as low (<6 stars), moderate (6–7 stars), or high (8–9 stars) quality. X * = X star.

**Table 3 healthcare-10-02204-t003:** ROBINS-I–The Risk Of Bias In Non-randomized Studies of Interventions assessment tool.

	Pre-Intervention		At Intervention	Post-Intervention				
	Confounding	Selection of Participants	Classification of the Intervention	Intended Interventions		Missing Outcome Data	Measurement of the Outcome	Selection of the Reported Result
Study				Effect of Assignment to Intervention	Effect of Starting and Adhering to Intervention			
Wan, 2017 [57]	Low risk	Low risk	Low risk	Low risk	Serious risk	Serious risk	Serious risk	Low risk
Chao et al., 2022 [60]	Moderate risk	Low risk	Low risk	Low risk	Low risk	Low risk	Moderate risk	Low risk
Gallagher-Ford et al., 2020 [50]	Low risk	Low risk	Moderate risk	Moderate risk	Serious risk	Low risk	Serious risk	Low risk
Hart et al., 2008 [51]	Serious risk	Low risk	Moderate risk	Moderate risk	Serious risk	Low risk	Serious risk	Low risk
Hsieh et al., 2020 [52]	Serious risk	Low risk	Moderate risk	Moderate risk	Low risk	Low risk	Moderate risk	Low risk
Mena-Tudela et al., 2018 [53]	Serious risk	Low risk	Moderate risk	Moderate risk	Serious risk	Low risk	Serious risk	Low risk
Moore et al., 2017 [54]	Low risk	Low risk	Low risk	Low risk	Low risk	Moderate risk	Moderate risk	Low risk
Moore et al., 2018 [55]	Low risk	Low risk	Low risk	Low risk	Serious risk	Serious risk	Serious risk	Low risk
Underhill et al., 2015 [56]	Moderate risk	Low risk	Moderate risk	Serious risk	Serious risk	Low risk	Moderate risk	Low risk
Waltz et al., 2022 [61]	Serious risk	Low risk	Moderate risk	Serious risk	Serious risk	Moderate risk	Moderate risk	Low risk

**Table 4 healthcare-10-02204-t004:** RoB 2.0: Revised Cochrane risk of bias tool for randomized trials–randomized trial methodologies.

	Randomization Process	Intended Interventions		Missing Outcome Data	Measurement of the Outcome	Selection of the Reported Result
Study		Effect of Assignment to Intervention (Part 1 and 2)	Effect of Adhering to Intervention			
Cardoso et al., 2021 [62]	Low risk	Some concerns Low risk	High risk	Low risk	High risk	Low risk
D’Souza et al., 2021 [64]	Low risk	Some concerns Low risk	High risk	Low risk	High risk	Low risk
Koota et al., 2021 [65]	Low risk	Some concerns Low risk	High risk	Low risk	High risk	Low risk
Levin et al., 2011 [8]	Some concerns	Some concerns Low risk	High risk	Low risk	High risk	Some concerns

#### 3.4.1. Multifaceted Educational Strategies Incorporating Mentoring and Tutoring

The different frameworks used in this category of interventions were the Advancing Research and Clinical practice through close Collaboration (ARCC) model, the Classification Rubric for EBP Assessment Tools in Education (CREATE), and the critical incident technique. The key component of the ARCC model is the central place of the mentor, an advanced practice nurse who assists nurses in honing their EBP skills and implementing projects to improve patient care and outcomes. Furthermore, the mentor actively participates in implementing strategies to overcome barriers within the healthcare environment by building a culture of EBP [68]. Guided by cognitive-behavioral theory, ARCC strategies are designed to encourage nurses to develop stronger beliefs about the value of EBP and more confidence in their ability to implement it consistently [67,69]. Levin et al. [8] proposed a multi-component ARCC model intervention program. The intervention started by clarifying the definition of EBP and the rationale for using it in clinical decision-making. In a second phase, the interventional group (IG) was taught how to explore searchable scientific databases to respond to clinical questions, and then participants were asked to find evidence by exploring appropriate scientific databases, registers, and websites. They were encouraged to explore the basic concepts of a systematic review method, especially reading and critically appraising meta-analyses. For 12 weeks, they had 1.5 h of focused follow-up with an EBP mentor to teach them how to implement the concepts and skills of EBP to address a clinical problem in daily practice. On the other hand, the control group (CG) received didactic content about adult physical assessment dispensed by an expert in EBP. Underhill et al. [56] investigated the beliefs and implementation of EBP after an institutional EBP education intervention that took place over 24 months. This consisted of four components, starting with face-to-face discussions with nurses to provide an introduction to EBP and EBP resources. The content of the EBP training was oriented toward quantitative research designs using the PICOT research question strategy (clinical population, intervention, comparison, outcome, time). It also included explanations of the levels of evidence, differences in quality improvement, nursing research and evidence-based definitions and methods. An overview of the content was presented on an EBP poster for guiding EBP projects (SPAWN projects). In 2012, a nursing scholarship day was organized to raise nursing staff’s awareness about the importance of providing evidence-based care. Finally, an online educational module on EBP was available. During each EBP session, conducted by a member of the Evidence-Based Practice and Innovation Committee who took on the role of mentor, participants could ask questions.

Gallagher et al. [50] evaluated the effects of a five-day EBP continuing education and skills-building program, which was attended by a total of 400 participants, including 377 RNs, between September 2014 and May 2016. The primary outcomes measured were: EBP attributes and competencies, beliefs, implementation, and knowledge using the ARCC model.

The final study to use the ARCC model was Singleton’s cohort design study [48]. It investigated the pre–post effects of an EBP teaching program using the EBP belief and implementation scales across seven years and five cohorts of the Doctor of Nursing Practice (DNP) program. The DNP program includes two education sessions on “Evidence-based Practice Methods and Techniques” involving 180 h of theory. Students also benefitted from a faculty mentor and a mentor from a clinical practice improvement project.

The second framework used was CREATE, a means of classifying EBP learner assessment tools. CREATE suggests that EBP knowledge should be assessed cognitively, using paper and pencil tests, as knowledge defines a learner’s retention of facts and concepts. Another assumption is that EBP skills should be assessed using performance tests, as skills reveal the application of knowledge. The assessment categories are: (a) Reaction to the Education Experience, (b) Attitude, (c) Self-Efficacy, (d) Knowledge, (e) Skills, (f) Behaviors, and (g) Benefit to Patients. The first three categories are assessed using self-reporting/opinion; knowledge and skills are assessed using performance testing; behaviors are tested using activity monitoring; and benefits to patients are assessed through patient-oriented outcomes [70]. Cardoso et al. [62] implemented an EBP educational program in a Portuguese nursing school in 2018. The educational program for the intervention cohort (IC) taught models for thinking about EBP and explained systematic review types, review question development, searching for studies, study selection processes, data extraction methods, and data synthesis. The last three sessions presented active methods through mentoring to groups of 2–3 students. The CG participants underwent their education as usual (theory, theory–practice, practice) delivered by the nursing educators.

Koota et al. [65] evaluated the effectiveness of an educational intervention on EBP for emergency nurses at two hospitals in Finland. The IG received the “Evidence-Based Practice Basics for Emergency Nurses” intervention, which included multifaceted educational strategies such as didactic lectures and discussions, small group tutorials, database search workshops, and a self-directed learning module. They also received support from the tutor teams, two emergency unit members (the researcher and a clinical nurse specialist), and a librarian for consultations. The CG completed a self-directed learning module entitled “Evidence-Based Practice Basics for Emergency Nurses”.

The study conducted by Mena-Tudela et al. [53] used the critical incident technique (CIT). The educational intervention was composed of two sessions: the first defined terms related to EBP and reflected on materials and using critical thinking, whereas the second involved practical exercises on information literacy. During their 12-week clerkship, student nurses had to identify a minimum of eight critical incidents, develop a PICO clinical question, and try to resolve the incident using a literature search. A lecturer was available to provide support and tutor feedback to groups of 8–10 students.

Different authors used the terms of mentor, EBP mentor, or faculty mentor [8,52,60,66], whereas others used tutoring or tutor teams. As Melnyk et al. stated [71], mentors work with nurses and are usually themselves advanced practice nurses (clinical nurse specialists or nurse educators) with superior knowledge and skills in EBP and organizational and individual behavior change. Additionally, some authors specified that mentors are opinion leaders within teams [7,21]. Mentoring involves a long-term relationship between two people, one of whom is usually older and/or more experienced than the other [72]. The aim of this relationship—based on mutual respect and compatible personalities—is to guide the mentee towards personal and professional growth [73]. Thus, a mentor’s role in an educational intervention is to provide support and social skills to the care teams involved and, more specifically, to assess how well they have acquired the skills taught. The tutoring described by Mena-Tudela et al. [53] was provided by nurse instructors, which is the traditional model of nurse tutoring [74]. Tutoring is a system of partnership throughout a learning process, placing the commitment and responsibility for the teaching and learning experience on the learner [75]. Even though the definitions and purposes of mentoring and tutoring are quite similar, tutoring requires the learner to be more proactive in the acquisition of new learning skills and does not require a one-to-one relationship, whereas mentoring does not always have a formal goal and does imply a one-to-one relationship.

#### 3.4.2. Single Educational Strategies

Three [55,58,59,61,65] of the five studies classified in this second category used a computer-based education (CBE) strategy, the fourth used face-to-face and videoconferencing sessions [61], and the fifth used the Promoting Action on Research Implementation in Health Services Model (PARiHS) and Plan-Do-Check-Act (PDSA) frameworks [57]. Using computer-based education in the teaching process [76] is an alternative method of providing information in a reliable, uniform manner [77]. In the Hart et al. [51] study, education on EBP was delivered through three computer-based learning modules. Module 1 helped students develop a clinical question, module 2 trained their ability to read and understand a research article, and module 3 demonstrated how to transform theoretical evidence into nursing practice. The studies by Moore [54,55] delivered an online educational intervention on EBP using Rogers’ Diffusion of Innovation Theory. The IG studied a computer-based module composed of a self-paced PowerPoint presentation addressing the relevance of EBP, the steps of EBP, the resources available to facilitate the implementation of EBP, and examples of EBP. Control group 1 was assigned a computer-based PowerPoint learning module on pain management, with a delivery format and time available similar to the intervention group. Finally, control group 2 underwent no intervention. Wan’s study [57] was guided by the PARiHS and PDSA frameworks. The EBP education program came in four units of 2–4 lessons each and was delivered through lectures with printed handouts, small group discussions, and questioning and writing exercises. The components were the EBP process, the PICOT model, literature search skills, fundamental research critiques, levels of evidence, quality improvement strategy, and the PDSA rapid cycle method. Finally, the educational intervention in the study by Waltz et al. [61] consisted of a practice and an interactive workshop (of 1.5 h each) to improve EBP competencies. These were conducted by Ph.D. nurse educators. The research topic was an example selected because of its relevance to their care environment.

#### 3.4.3. Multifaceted Educational Strategies Using the Five Steps of Evidence-Based Practice

The last three studies [56,64,67,70] retrieved were guided by the five basic steps of EBP: (a) Ask a question; (b) Search for information or evidence to answer the question; (c) Critically appraise that information or evidence; (d) Integrate the appraised evidence using clinical expertise and patients’ preferences; and (e) Evaluate [78]. The intervention proposed by Hsieh and Chen [52] consisted of an evidence-based school-nursing program that used multiple pedagogical strategies, such as a traditional in-class format, a combination of teaching strategies, a flipped classroom format, group activities, case studies and supervision, mobile learning technology, and online lessons. The RCT conducted by D’Souza et al. [64] involved a 30 h EBP training intervention over four days at a nursing education institution. Different strategies were used, such as lectures and discussions, small-group activities, critical-thinking exercises, scenario-based discussions, online library training, brainstorming sessions, critical appraisal sessions, activities on integrating EBP, EBP booster sessions, or sessions on literature searches. Finally, Chao et al. [60] developed an intervention consisting of two pedagogical strategies: a flipped e-learning intervention and traditional classroom teaching.

### 3.5. Clinical Outcomes, Instruments, and Results

#### 3.5.1. Primary Outcomes

The 15 studies measured 11 primary outcomes—EBP beliefs, EBP self-efficacy, EBP attitudes, perceived EBP implementation, EBP competencies, EBP knowledge, EBP skills, EBP behaviors, EBP desire, EBP practice, and perceptions of organizational culture and readiness (EBP attributes)—using 15 qualitative and quantitative instruments.

(i)Evidence-Based Practice Belief Scale (EBP-B)

Six studies [8,52,54,60,61,69] used this instrument. In their longitudinal, pre-experimental study, Gallagher et al. [50] initially showed increasing mean scores from T0 to T1 (T0: M = 58.4 (SD = 8.7), T1: M = 68.1 (SD = 8.1)), and then decreases from T1 to T2 (M = 66.3 (SD = 9.1) and increases from T2 to T3 (M = 68.1 (SD = 7.4)). These scores were statistically significantly different between T0 and T1 (*p* < 0.001), between T0 and T2 (*p* < 0.001), and between T0 and T3 (*p* < 0.001).

In the study by Underhill et al. [56], oncology RNs and advanced practice nurses completed this quantitative tool at T1 and T2. The EBP-B median at T1 was 56.5 (range = 37–77, IQR1–3 = 50–61) and at T2 was 57 (range = 38–76, IQR1–3 = 51–63). There were no significant differences between T1 and T2 implementation scores or between those who had or had not participated in the SPAWN projects (*p* = 0.36). Finally, the level of nursing education was positively correlated with EBP-I (r = 0.32, *p* = 0.01), unlike years employed as a nurse (*p* = 0.16). The study by Levin et al. [8] demonstrated a major effect for its IG (F[1,15] = 10.39, *p* = 0.006), with nurses showing better results in EBP at T3 and T4. There was also a main effect of time (F[2,30] = 5.85, *p* = 0.007) since those authors noted a significant increase from T1 to T3 (T1: M = 12.89 and T3: M = 28.14; F[1,15] = 12.40, *p* = 0.003). Finally, there was also a significant time–group status interaction (F[2,30] = 3.625, *p* = 0.0039) since the IC showed a higher mean score on EBP implementation than the CG (M = 15.50 versus 10.33). These results were consistent with different studies [57,65]. Indeed, the CG’s emergency nurses revealed constant attitudes toward EBP throughout their follow-up. Nurses from the IG showed improved attitudes toward EBP at T2, with a decrease at T3 (M = 4.3 (SD = 0.23) versus M = 4.12 (SD = 0.49)), although still higher than at T0 (M = 3.85 (SD = 0.49)).

(ii)Perceived Evidence-Based Practice Implementation (EBP-I)

Five studies used the EBP-I, whereas two studies used the Adapted EBP-I. In Singleton’s study [48], DNP–FNP students showed a mean pre-intervention score of 2.68 (SD = 0.94) versus 3.61 (SD = 0.96) post intervention. Data analysis showed this to be statistically significant (t = 8.4 (52), *p* < 0.001), with an effect size ranging from 0.75 to 1.5 SD above the mean. Finally, gains were made by every cohort and across the curriculum (F ratio 20.01). Underhill et al. [56] asked oncology RNs and advanced practice nurses to complete the EBP-I at T1 and T2. No statistical differences were found between the scores of nurses who had and had not participated in SPAWN projects (*p* = 0.36). However, nurse leaders had higher perceived scores (*p* < 0.001). Finally, the level of nursing education was positively correlated with EBP-I (r = 0.32; *p* < 0.01). These results are consistent with Wan’s study [57]. Nurses in the IG who underwent the EBP education program had lower mean rank scores than CG nurses in the pre-test but higher ones in the post-test. Their scores increased between T0 and T1 (*p* = 0.025), unlike the CG (*p* = 0.257). Mean scores across the 18 implementation items used by Gallagher et al. [50] increased over time, although no data were reported for T1 (T0: M = 15.4 (SD = 13.9), T2: M = 20 (SD = 12.7), and T3: M = 25.4 (SD = 15.6)). Statistically significant differences were found between T0 and T2 (*p* = 0.0087) and between T0 and T3 (*p* > 0.0001). These results are consistent with Levin et al. [8]. The interventional group showed significantly higher scores from T1 to T3 (T1: M = 12.89, T3: M = 28.14) (F[1,15] = 12.40, *p* = 0.003). However, the study by Koota et al. [65] revealed that the IG’s emergency nurses’ EBP scores had improved at T2 but had decreased again by T3. Some studies showed conflicting findings on how EBP behaviors decreased following interventions [54,55], and the EBP scores of nurses in the CG had dropped below baseline levels by T3.

(iii)Evidence-Based Competencies Scale

Gallagher et al. [50] assessed nurses’ EBP competencies using the Evidence-Based Competencies Scale. Participants completed the scale at four time points, and mean scores for the 24 competency items increased over time (T0: M = 53.1 (SD = 18), T1: M = 75.7 (16.3), T2: M = 77.3 (SD = 17.7), and T3: M = 85 (SD = 19.9)). Statistically significant differences were found between T0 and T1, T0 and T2, and T0 and T3 (*p* < 0.001).

(iv)Evidence-Based Practice Knowledge Assessment Questionnaire (EBP-KAQ)

The EBP-KAQ was developed in 2015 (Gallagher-Ford et al.) but remained unpublished. Gallagher et al. [50] reported mean scores for the 38 EBP-KAQ questions increasing over the four time points (T0: M = 24 (SD = 6.1), T1: M = 30.2 (SD = 3.9), T2: M = 31.2 (SD = 3.5), T4: M = 31.7 (SD = 4.2)). Statistically significant differences were found between T0 and T1, T0 and T2, and T0 and T3 (*p* < 0.0001), providing strong evidence of a positive effect on participants.

(v)Adapted Fresno Test

Two studies used the Adapted Fresno test to evaluate the knowledge, skills, and EBP competency of nurse educators. In the study by Cardoso et al. [62], both the IG and the CG completed the Adapted Fresno Test at T0 and T1. The mean IG score improved from baseline (M = 6.85 (SD = 5.15)) to T1 (M = 12.47 (SD = 7.21)), as it did for the CG (T0: M = 7.25 (SD = 5.34), T1: M = 9.73 (SD = 5.56)). These improvements over time were statistically significant for both the IG and the CG (F(1.73) = 53.028, *p* < 0.001, and F(1.73) = 13.832, *p* < 0.001). Finally, students in both groups significantly improved their knowledge and skills in four items (1, 3, 4, 5), only IG improved in item 7 (Z = −2.543, *p* = 0.011 versus CG: Z = −1.941, *p* = 0.052), and only CG improved in item 7 (Z = −2.714, *p* = 0.007 versus IG: Z = −1.236, *p* = 0.216). D’Souza et al. [64] asked nurse educators to complete the Adapted Fresno Test at T0, T1, and T3. The mean IG score improved from T0 to T2 (M = 20.30 (SD = 13.13) to M = 103.45 (SD = 27.87)), whereas mean CG scores only improved a little (T0: M = 24.67 (SD = 11.39) and T2: M = 28.78 (SD = 10.86)). The differences between the IG and CG indicated that the EBP training program improved EBP competency (F[2,92.06] = 37.13, *p* < 0.05).

(vi)Evidence-Based Practice Questionnaire-FI (EBPQ-FI) or Evidence-Based Nursing Questionnaire

Koota et al. [65], D’Souza et al. [64], Wan [57], and Moore [54,55] all used the EBPQ. Hart et al. [51] used the same instrument but named it the Evidence-Based Nursing Questionnaire.

In Moore’s studies [54,55], RNs rated themselves moderate in attitudes to EBP (M = 5.28 (SD = 1.02)) and low in use (M = 4.80 (SD = 1.23)) and knowledge and skills (M = 4.83 (SD = 0.99)). Koota et al. [65] used the EBPQ to assess behaviors, attitudes, knowledge, and skills at four time points. At T2, the IG showed statistically significant improvements in EBP attitudes (*p* = 0.019) and knowledge (*p* = 0.005). Meanwhile, EBP behaviors had decreased among the CG. At T3, there was a statistically significant difference between the groups’ EBP attitudes (*p* = 0.010). EBP attitudes remained constant for the CG across the whole follow-up period, with an improvement at T2 but then a decrease at T3. The IG showed statistically significant improvements in their knowledge of EBP at every time point (*p* < 0.001, *p* < 0.001, and *p* < 0.001). Finally, the two CGs showed lower scores for EBP behaviors, whereas they improved at T2 and then decreased at T3 for the IG. Finally, the IG’s EBP skills showed a minor improvement at T2 (T0: M = 4.64 (SD = 0.76), T2: M = 5.05 (SD = 0.42)). In the study by Wan, the IG had lower mean ranks for attitudes to EBP than the CG in the pre-test scores but higher ones in the post-test. The IG did not increase attitude scores between T0 and T1, as CG did. D’Souza et al. [64] used the EBPQ to assess knowledge and use of EBP. Both groups’ knowledge increased gradually from T1 to T2. The 30 h EBP training intervention was effective in improving nurse educators’ knowledge (F[2,91.65] = 4.11, *p* < 0.05). After T1, the IG showed a greater increase in EBP use scores than the CG. IG nurse educators showed statistically significant improvements in their EBP use scores (F[2,94.88] = 6.21, *p* < 0.05). RNs in the Hart et al. [51] study completed the Evidence-Based Nursing Questionnaire before and after computer-based education sessions. Statistically significant pre- and post-intervention differences were found for knowledge of EBP (t (0.05, 312) = –2.296, *p* = 0.022), attitudes to EBP (t (0.05, 313) = –4.750, *p* < 0.001), skills in EBP (t (0.05, 313) = –4.322, *p* < 0.001), and organizational readiness for EBP (t (0.05, 313) = –8.601, *p* < 0.001) and its use in research.

(vii)Evidence-Based Practice Competence Questionnaire (EBP-COQ)

Mena-Tudela et al. [53] used the EBP-COQ to evaluate self-perceived levels of EBP competence among Spanish nursing students. Nearly all the items showed significant differences between the three measurement points, except for item 1 (*p* = 0.099), item 4 (*p* = 0.051), item 7 (*p* = 0.055), item 10 (*p* = 0.065), and item 11 (*p* = 0.441). Finally, statistically significant overall differences between the three time points were measured (*p* < 0.001), between T0 and T1 (*p* < 0.001) and between T0 and T2 (*p* < 0.001).

(viii)School Nurse Evidence-Based Practice Questionnaire (SNEBP)

Hsieh and Chen [52] used the SNEBP at three different time points. Knowledge section scores had increased by an average of 0.75 at post-training (*p* < 0.001) and by 0.37 at follow-up (*p* < 0.001). Self-efficacy section scores had increased by an average of 0.3 at post-training (*p* < 0.001) and 0.13 at follow-up (*p* < 0.001). Skills section scores significantly increased from baseline (M = 3.73; SD = 0.58) to post-training (M = 3.80; SD = 0.62) (*p* < 0.001), but they then significantly decreased (by an average of 0.1) between post-training and follow-up (*p* > 0.005). Finally, attitude section scores made no statistically significant changes over time.

(ix)Organizational Culture and Readiness for System-Wide Integration of Evidence-Based Practice Scale

Gallagher et al. [50] used the Organization Culture and Readiness for System-Wide Integration of Evidence-Based Practice Scale. This assesses cultural factors that can influence system-wide implementation of EBP and overall perceived readiness for EBP integration. The mean score for the 24 competency items increased across the three time points as no values were reported for T1 (immediately after the intervention) (T0: M = 77 (SD = 16.1), T1: M = 75.7 (16.3), T2: M = 80.4 (SD = 15.4), or T3: M = 84.6 (SD = 15.9)). Statistically significant differences between T0 and T2 (*p* = 0.0041) and between T0 and T3 (*p* > 0.0001) were described.

(x)Twenty-one-item scale

D’Souza et al. [64] developed a 21-item for their study in order to respond to the particularities of their population and assess their attitudes to EBP. Attitude scores were higher in the IG across time points T1 and T2. Significant changes in the attitudes of the IG nurse educators (F[2,91.07] = 3.55, *p* < 0.05) were described.

(xi)Thirty-item questionnaire

Chao et al. [60] used a 30-item questionnaire adapted from Lee et al.’s (2011) study [79]. It included the three dimensions of knowledge, attitudes, and practice. For the knowledge subscale, there was no significant difference in the two groups’ mean scores at T1 and T2 (*p = 1.00* and *p* = 0.702). Both groups showed improved knowledge after the intervention. Both groups’ attitude scores (CG: *p* = 0.000; IG: *p* = 0.000) and practice scores (CG: *p* = 0.000; IG: *p* = 0.000) also improved after the intervention, and there were no differences between the two groups at the three different time points (T1: *p* = 0.573; T2: *p* = 0.792; T3: *p* = 0.153).

(xii)Healthcare EBP Assessment Tool (HEAT)

Waltz et al. [61] used HEAT to measure self-perceived competencies in EBP, including understanding, ability, desire, frequency, and barriers before and after the intervention. This consisted of an interactive workshop of 1.5 h each. The pre-test and post-test scores showed improvements for the EBP subscales of frequency (except for “Used research findings in practice”: *p* = 0.179), ability (except for “Participate in a research project”: *p* = 0.066), desire, and barriers (only “I have difficulty finding research or library materials with my searches” and “I have difficulty understanding research articles” were statistically significant; *p* = 0.019 and *p* = 0.035, respectively).

(xiii)Qualitative analysis of the monographs

Cardoso et al. [62] qualitatively analyzed 18 students’ monographs using an evaluation form with 11 guiding criteria (review questions, inclusion and exclusion criteria, methodology, the presentation of results, and the congruence between the review questions and the conclusion’s answers). The CG’s monographs were of a lower quality than the IG’s since the CG displayed difficulties synthesizing the data and providing clear answers to the review questions.

(xiv)Tailored satisfaction questionnaire

Underhill et al. [56] aimed to explore barriers to and facilitators of implementing EBP. The most common barriers experienced by participants were a lack of time, knowledge, and access to online journals or databases. That is why oncology nurses requested more time, resources, education, and awareness about EBP. On the other hand, the facilitating factors identified were nursing leadership and leaders from nursing research. Chao et al. [60] also used a questionnaire to discover levels of satisfaction with the learning process and course, with ratings from 1 (strongly disagree) to 5 (strongly agree). Higher scores indicated greater satisfaction in the learning process and course. There were no significant differences in the scores between the two groups (*p* = 0.001; CG: M = 38.49, SD = 0.99 and IG: M = 44.24, SD = 0.60). The IG showed significantly higher scores for each item and overall satisfaction than the CG. The item with the lowest score for the CG was “I could understand EBP through the course content” (M = 3.75, SD = 1.15), and the highest score was for the item “Practical experience improved my ability to search for empirical literature” (M = 3.96, SD = 0.96). The item with the lowest score for the CG was “The workshop helped me complete the EBP 5As and apply them to clinical care” (M = 4.35, SD = 0.52). The most favorably rated items were “I could understand EBP through the course content” (M = 4.46, SD = 0.66) and “The course strengthened my ability to search for empirical literature (M = 4.46, SD = 0.50).

#### 3.5.2. Secondary Outcomes

Only two studies evaluated secondary outcomes, but they used eight instruments. Levin et al. [8] used a qualitative and/or quantitative approach to deepen their investigation of five secondary outcomes: group cohesion, job satisfaction, nurse productivity, nurse attrition/turnover, and professionalism and leadership. Finally, Koota et al. [65] examined learner satisfaction using instruments developed by those authors specifically for their study.

(i)Group Cohesion Scale

Levin et al. [8] used the Group Cohesion Scale developed by Good and Nelson in 1973 [80]. Scores for group cohesion were examined according to time point and group as well as time point–group interactions. No main effects were identified for group (F[1,16] = 0.016, *p* = 0.502), time point (F[2,32] = 2.50, *p* = 0.098), or time point–group interactions (F[2,32] = 0.824, *p* = 0.448). However, the IG did report better group cohesion at T3 than at T1 (F[1,21] = 5.580, *p* = 0.028).

(ii)Index of Work Satisfaction (IWS)

Levin et al. [8] use the IWS to measure work satisfaction according to time point, group, and time point–group interaction. Statistical analyses showed no main effects for group (F[1,22] = 0.057, *p* = 0.814) or time point–group status interaction (F[1,31] = 0.392, *p* = 0.537). However, it did show a main effect for time (F[1,22] = 0.016, *p* = 0.900).

(iii)Visiting Nurse Service of New York

Levin et al. [8] assessed the nurse productivity of the VNSNY. No main effects for group (F[3,52] = 0.422, *p* = 0.738) or time point (F[1,52] = 2.799, *p* = 0.100) were described. The overall mean productivity rate at T1 (M = 4.59) did not differ significantly from the rate at T3 (M = 4.59 versus M = 4.77). Finally, no significant time point–group status interaction (F[3,52] = 0.243, *p* = 0.866) was reported.

(iv)Visiting Nurse Service of New York–Human Resources

Levin et al. [8] used the VNSNY–Human Resources database to evaluate nurse attrition. In the IG, attrition rates decreased from 11% (*n* = 5) in 2004 to 5.7% (*n* = 3) in 2005, whereas the attrition rate in the CG was 35%.

(v)Focus group

Levin et al. [8] conducted focus groups to explore nurse managers’ and visiting staff nurses’ professionalism and leadership. No more details were given. These exchanges with study participants highlighted their greater sense of professionalism and increased ability to see managers as colleagues.

(vi)Learning questionnaire

Levin et al. [8] compared the intervention’s learning effects on the IG and the CG at T1 and T2 via a learning questionnaire. The IG answered more EBP questions correctly than the CG, but differences were not statistically significant.

(vii)Knowledge test and course assessment form

Koota et al. [65] developed these instruments because they could find no validated tools appropriate for their study. Groups differed significantly in their satisfaction with the teacher’s ability to encourage nurses to ask clinical questions (IG: M = 2.98 (SD = 0.62) versus CG: M = 2.75 (SD = 0.93)) (*p* > 0.001). In addition, groups significantly differed in their satisfaction with the usefulness of considering research evidence in nursing practice (IG: M = 3.65 (SD = 1.06) versus CG: M = 2.94 (SD = 0.83)) (*p* = 0.01).

## 4. Discussion

In increasingly complex healthcare systems, and given the inherent challenges of healthcare environments, it is becoming essential for nurses to have a solid foundation in EBP because of its proven links to improved patient and organizational outcomes [61]. This systematic review identified and summarized the most effective educational interventions to increase skills in EBP among nurses. It identified three types of key educational interventions: multifaceted educational strategies incorporating mentoring and tutoring, single educational strategies (often delivered online), and multifaceted educational strategies using the five basic steps of EBP. These pertinent findings will provide tangible support to researchers, clinicians, healthcare system managers, healthcare institutions, health policymakers, stakeholders, universities and their faculty members, and healthcare professionals because they reveal an assessment of the effectiveness of different educational interventions on eleven primary outcomes. The review covered studies involving 2712 RNs and licensed practical nurses (LPNs), bachelor’s degree students, Ph.D. students in Nursing Practice–Family Nurse Practice, nurse educators, emergency nurses, nurse managers, and visiting staff nurses.

EBP educational interventions should be a standard part of nurses’ professional development in clinical settings. Healthcare systems and nurses have been encouraged to adopt the principles of EBP in order to improve the delivery of safe, high-quality healthcare to patients. We might expect EBP to be well known and understood by nurses—and equally prevalent in their education programs and clinical practice—but although they generally have positive beliefs about EBP, many individual and organizational barriers remain [48], impeding them from turning their beliefs into action. These barriers include the lack of time, deficiencies in knowledge about EBP, access to evidence, autonomy, empowerment to change practice, resistance from colleagues or managers [57,60,68], and even inadequate numbers of EBP mentors [50]. To ensure EBP is used daily, it is necessary to identify the most efficient and effective EBP educational program to improve EBP beliefs, EBP self-efficacy, perceived EBP implementation, EBP competencies, EBP knowledge, EBP skills, EBP attitudes, EBP behaviors, EBP desire, EBP practice, and perceptions of organizational culture and readiness, to allow individuals, leaders, and healthcare organizations to build a culture and environment of EBP, and to make EBP’s adoption sustainable [35].

“EBP knowledge and skills can be acquired through formal and continuing education and self-paced online tutorials” [81]. Indeed, the 15 studies included in this systematic review present different strategies to implement the interventions described. Although the three different types of interventions identified had significant positive impacts on the 11 outcomes mentioned above, EBP cannot occur in isolation: it requires early adopters, champions, and teamwork. According to our findings, the best way to increase EBP skills among nurses and facilitate EBP implementation in their daily practice seems to be the use of a variety of educational strategies that integrate multiple learning pathways and techniques with regular follow-up and feedback from mentors and/or leaders. Computer-based education (CBE) currently appears to be the most cost-effective and efficient strategy, providing reliable, consistent delivery of information and the flexibility for caregivers to choose where and when to access the intervention. This type of educational strategy ensures that EBP’s adoption by nurses is sustainable in their daily practice.

Not all RNs have undergone training on EBP during their educational or career trajectory. Thus, it is important to develop educational strategies for both stages so as to standardize and homogenize EBP knowledge and skills: (1) during their student curricula, and (2) regularly, on site, where they practice. Indeed, mere theoretical input is not enough, and educational interventions in clinical settings are essential to promoting the use of EBP. The literature concludes that nurses’ EBP competencies are only low to moderate internationally, particularly in terms of their EBP knowledge, EBP skills, and their confidence in employing EBP. One potential explanation is the heterogeneity in the quality and content of EBP teaching programs in students’ curricula [29]. It is therefore important to homogenize nurses’ knowledge and skills while they are still in training. Only once nurses are competent in EBP will they be more likely to engage in EBP in their daily practice [11].

In addition to using the different strategies mentioned above, researchers must also examine the variables that might support or hinder the implementation of EBP, using reliable and valid measures [7,82]. In this context, the Consolidated Framework for Implementation Research (CFIR), developed by Damschroder et al. [82,83], sheds light on a wide range of factors that can support or hinder implementation. The CFIR is itself the result of a literature review; it synthesizes all the existing implementation models, bringing together the elements facilitating or constraining the implementation of new practices described in the literature. It is subdivided into five areas: (i) the external environment, which analyzes, among other things, cosmopolitanism; (ii) the internal environment, which looks at structural characteristics and culture; (iii) the personal characteristics that influence the course of a project; (iv) the characteristics of the intervention; and (v) the implementation process, at both the individual and organizational levels. According to its developers, the CFIR can be used at three time points. Firstly, during pre-implementation, to conduct a baseline survey (nurses’ sociodemographic characteristics, such as level of education, were positively correlated with EBP-I (r = 0.32, *p* = 0.01) in the studies by Gallagher et al. [50], Underhill et al. [56], and Wan [57]) including the medical specialty’s specific culture, leadership style, openness to change, learning climate, communication about the project, and an identification of its implementation process. Educational interventions are only one part of advancing EBP. Indeed, implementing EBP is also affected by nurses’ demographic characteristics, such as age [84], marital status [85], years of work experience, or educational level [86], as well as their learning environment [87], staffing levels, and leadership styles [88]. Secondly, during peri-implementation, CIFR can be used to enable monitoring (assessing the characteristics of an intervention, such as its clarity or adaptability, adapting the implementation process through PDSA cycles, regular communication, and feedback, responding to the specialty’s needs, and mobilizing the right leadership style). Thirdly, it can be used post implementation to ensure sustainability—including continuous adaptation based on feedback and active involvement from the specialty, regular communication, and the identification of mentors and/or opinion leaders [7,17]—and avoid a drop in buy-in as was observed in the study by Koota et al. [65]. In addition to the CFIR, there are other models that can guide the implementation of EBPs, such as Promoting Action on Research In Health Services Framework (PARiHS) [89], the Iowa Model of Evidence-Based Practice to Promote Quality Care [90], Advancing Research and Clinical practice through close Collaboration (ARCC) model [91], the JBI Model of Evidence-Based Healthcare [92], Johns Hopkins Nursing Evidence-Based Practice, or even the Clinical Scholar Model [93].

### 4.1. The Review’s Limitations

There are some limitations to this review that should be considered. First, despite a thorough literature search using recognized methodological guidelines and recommendations, and the use of many different terms to describe EBP, our review may have missed some studies that met all the selection criteria, either due to study search errors or investigators’ mistakes. Second, language and publication bias may be present despite the review’s scope being worldwide. Third, the effects of different types of healthcare system, systems of professional training, urban or rural settings, and socioeconomic factors, among others, were not analyzed. Finally, this review originally aimed to answer the question, “How effective are educational interventions to increase skills in EBP among nurses and physiotherapists delivering primary healthcare?” Despite a literature search that took place at three distinct time points (2019, 2021, and 2022), no relevant studies involving PTs were found. The study’s results can therefore only be generalized in relation to our sample’s heterogeneous nursing characteristics (RNs, LPNs, bachelor’s degree students, Ph.D. students in Nursing Practice–Family Nurse Practice, nurse educators, emergency nurses, nurse managers, and visiting staff nurses). However, the Gallagher-Ford et al. study goes beyond this sample to include 23 healthcare professionals [52]. Nevertheless, one of the review’s strengths is that it included four RCTs, despite their poor methodological quality, eight quasi-experimental studies, and one cohort study, all of which used validated instruments when measuring study outputs. Furthermore, we used highly recommended methodological norms and guidelines, which render our findings extremely reliable.

### 4.2. Recommendations

In view of the small number of studies published in our field of interest to date, future research should consider whether the present findings are applicable to other samples. Indeed, there is a need to include other primary HCPs to identify which interprofessional educational interventions will ensure that they too can provide the best available evidence-based healthcare. With a view to ensuring continuous improvements in safe, high-quality healthcare for patients, more research data on the most efficient, effective educational interventions for meeting HCPs’ needs and ensuring that EBP’s added value is delivered at the patient’s bedside also need to be collected. More data will also help to establish guidelines for EBP implementation. Finally, evidence from more high-quality studies involving larger sample sizes of HCPs would be welcome to test education interventions on EBP.

Our recommendations for implementing EBP in clinical practice and ensuring that implementation succeeds require a consideration of the specific characteristics and needs of the caregivers involved, as well as the uniqueness of the clinical setting, medical specialty, and the educational intervention before, during, and after implementation. Indeed, effective implementation of EBP comes through repeated exposure to and practice of EBP. This systematic review’s findings emphasize that strategic leadership, communication, and teaching strategies for EBP should rely heavily on the added value created by the mentors/tutors who informally and/or hierarchically take clinical leadership among their peers. However, there is a need to clearly identify the facilitators of and barriers to implementing educational interventions in healthcare settings. Our recommendations for future research also include analyzing and comparing the factors facilitating and constraining the pre-, peri-, and post-implementation phases of an educational intervention about EBP. This will help RNs, other HCPs, and nurse educators to adjust and improve their knowledge of implementation processes. That future research should use the CFIR during all three phases. It would also be important to consider whether different types of professional training, urban or rural settings, or socioeconomic factors had any impact on the effectiveness of educational interventions. This could be carried out during the CFIR’s pre-implementation phase.

## 5. Conclusions

To the best of our knowledge, this systematic review is the first on this topic to combine samples of RNs and LPNs, bachelor’s degree students, Ph.D. students in Nursing Practice–Family Nurse Practice, nurse educators, emergency nurses, nurse managers, and visiting staff nurses. Nurses corroborated the importance of evidence-based practice (EBP), but very few of them were able to implement it due to personal (lack of confidence, knowledge, and time, plus heavy workloads) and organizational (lack of material, human resources, support from the hierarchy, and implementation models) barriers. Systematically integrating best evidence into daily practice is challenging due to the inherent complexity of the EBP implementation process.

In conclusion, promoting the appropriate organizational culture and continuously developing the infrastructure, resources, and administrative support needed to advance the use of EBP is itself a process of moving from evidence to practice. However, this is not enough. Indeed, this paper highlighted the significant influence that educational strategies can have on the ability to master and implement EBP. The findings demonstrated that appropriate educational strategies could help EBP become a standardized part of the nursing student curriculum and of RNs’ daily practice. The most efficient, effective educational interventions will meet professionals’ needs and ensure that EBP’s added value is delivered at the patient’s bedside. These ongoing interventions seem to be relevant to continuing education based on multifaceted learning strategies with regular follow-up and feedback from mentors and/or leaders. Computer-based education is the most cost-effective and efficient strategy.

## Figures and Tables

**Figure 1 healthcare-10-02204-f001:**
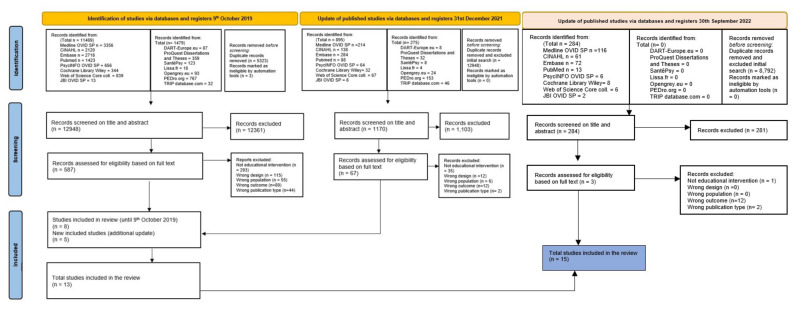
Search strategy for the EDITcare systematic review [46].

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
