# Peer review of "Effectiveness of Educational Interventions to Increase Skills in Evidence-Based Practice among Nurses: The EDITcare Systematic Review"

_healthcare, 2022, doi:10.3390/healthcare10112204_

Round 1

Reviewer 1 Report (Previous Reviewer 2)

Dear authors, 

In my view you have addressed all the recommendations and suggestions I made in the first review. So, congratulations on the job. Now I'm just suggesting revising Table 1 because some words should be better aligned (e.g. in Study 1, in the sample section there is too much space between "-" and "Middle Age")

Best regards, 

Author Response

Reviewer 2 Report (New Reviewer)

1.What assumptions authors made during the simulation phase of this research work? Provide a critique on this aspect.
2.Background information of Evidence-Based Practice among nurses to be provided.

3.Table 1 to be aligned  in the proper format.

4.The Limitations of the proposed study need to be discussed before conclusion.

5.The authors should improve their literature review by including recently published papers.

6.I feel that more explanation would be need on how the proposed method is performed.

7.identified research gaps and contribution of the proposed study should be elaborated.

8.What is the motivation of the proposed work? Research gaps, objectives of the proposed work should be clearly justified.

9. The manuscript should be deeply revised for English correctness and readability should be improved. As there is some grammatical issues and lingual mistakes which needs to be fixed.

Round 2

Reviewer 2 Report (New Reviewer)

1.The motivation of the paper needs to be more clear as the authors only list what have they done but the reason why is it required is not mentioned.

2.There is need of more precise and authentic sources of information for the facts presented in the paper related to educational interventions to increase skills in  Evidence-Based Practice among nurses

Author Response

I would like to sincerely thank you for your relevant feedback, which helped to improve the quality and relevance of our research.

However, would it be possible to clarify your comment number 2: "There is need of more precise and authentic sources of information for the facts presented in the paper related to educational interventions to increase skills in Evidence-Based Practice among nurses". Data on EBP educational interventions are drawn from the 15 articles selected for the systematic review, and the concept of EBP is discussed in relation to the definitions of the Rapports "To Err is Human" and "Crossing the quality Chiasm", and to the differents authors who have developed and studied this concept.

Thank you for your clarification

This manuscript is a resubmission of an earlier submission. The following is a list of the peer review reports and author responses from that submission.

Round 1

Reviewer 1 Report

1. In Abstract section, please add more information to Background (line 16).

2. Description of section 2 may make readers confuse. I suggest the authors should re-organize the section 2 to explain (1) what/how the materials are selected (e.g. based on what criteria)? (2) how to collect and process the collected materials? The authors should distinguish and describe based on these two directions to make the statements more specific and clearer. The description based on the current sub-sections is not clear and somewhat messy.

3. I suggest that Limitations and Recommendations (from line 676-701) should be moved to Conclusions section.

4. The manuscript should be proof read, otherwise it is difficult to understand and read. Please re-check the grammar and spelling.

Reviewer 2 Report

Dear authors,

Thank you for the opportunity to review your interesting and well written manuscript. I will give my feedback following the structure of the manuscript. 

1.Title and abstract

1.The title is informative and the abstract provides a summary of the manuscript's major aspects. I also have a doubt. In the abstract the authors say that no relevant studies were found in physiotherapists but when you read the title you hope to find some interest results closer to the physiotherapists. Wouldn’t it be better to redefine the title?

2.Introduction

After reading the introduction (first paragraph) it seems that the authors only consider EBP important for professionals working in the hospital, then both primary health care and their professionals are introduced. I would like to read about the importance of EBP for all the professionals in different health settings during all the introduction.

All articles in the background are from more than 5 years ago. Some from more than 10 years. I suggest authors to update this chapter.

3.Materials and Methods

Under my point of view, the material and methods chapter is very well written, providing all the information that the readers need. I would congratulate the authors for this part of the article.

4.Results

The results are clear, well-structured and well written, especially the instruments part. I would congratulate the authors for this chapter.  I only recommend the authors to review table 1, because it is quite difficult to read. In point 3.3.1, I would like to read the difference between mentoring and tutoring, sometimes these concepts could be exchanged. The results are presented in different order in the abstract and in the results chapter. I recommend the authors to homogenize it.

5.Discussion and conclusions

Discussion begins with a summary of the main objectives and then well discussed their results with the current literature. Conclusion chapter is clear and well written, answering the objectives of these study. So I have nothing to add in these two chapters.

Reviewer 3 Report

First of all, congratulations on the initiative and for the work done. Physiotherapy is mentioned in the title, but then it is indicated that no relevant studies about educational interventions in EBP among physiotherapists were found.

Regarding this review, its objective is to assess the effectiveness of Evidence-Based Practice educational interventions among health professionals who assess primary health care.

For years, the importance of scientific evidence in care has been demonstrated, as already indicated in the introduction. It must be taken into account that the initial search proposed provides a high number of articles (12,948), including 587 for full-text reading and analysis. In this sense, there are two temporal moments for the search (October 2019 and December 2021) and only 14 are included.

The study protocol is published on 11/2/2020 at https://www.researchprotocols.org/2020/11/e17621/ and submitted on December 27, 2019 after the search (https://preprints.jmir. org/preprint/17621?__hstc=178719527.bb8fc8f7c8a7069039146824545d0306.1657022859320.1657022859320.1657022859320.1&__hssc=178719527.2.1657022859320&__hsfp=384538321).

Surprisingly, the effects of the type of health system, the training of professionals, the urban or rural area, and socioeconomic level, among others, are not analyzed in the article, considering that 8 articles are from the USA and 6 from different countries. On the other hand, studies with diverse methodological designs are included, indicating in the results that they are of poor methodological quality or indicating "Finally, the RCT studies were scored as "of some concern" regarding their methodological risk of bias." Do you consider it appropriate to have included works that show research designs of low methodological quality? Can the results of studies with different methodological quality be considered comparable in this way? Have you included any strategy to control this methodological bias?

Regarding the results, sometimes it is limited to describing the different aspects seen in the articles (description of the individual intervention, etc.). On the other hand, regarding the results, 10 primary outcomes are assessed using different tools. From there, he describes the literal results of each study in which it has been used. It is indicated that they are validated tools but they do not indicate how the instruments have been validated.

On the other hand, it indicates "Secondary Outcomes" when only two articles indicate it and use different instruments.

Regarding the discussion, it seems more like a synthesis of the results itself than a discussion of the importance of practice based on scientific evidence. On the other hand, as the authors indicate, the review presents a series of limitations in the search and analysis of the articles (interobserver variability, critical reading tools, etc.). The conclusions are obvious regarding the importance of implementing scientific evidence in training at all levels: This systematic review demonstrates the need for EBP to become a systematic part of the nursing student curriculum, followed up by relevant continuing education on EBP that mobilizes support and leadership, raises awareness about EBP, implements it, and ensures its sustainability. However, it does not respond to the objective of the article (How effective are educational interventions to increase skills in EBP among nurses and physiotherapists delivering primary healthcare?

Round 2

Reviewer 3 Report

The document has improved with revisions. The change of title and the improvement of the introduction, allow a better understanding of the text. The limitations explained in the text must be taken into account. The inclusion of studies with poor methodological quality continues to draw attention, as does the inclusion of diverse populations (nursing students, emergency room nurses, etc.). However, I continue to see deficiencies in the discussion (short and not responding to the objectives), as well as in the conclusions.
